# Identification of factors directly linked to incident chronic obstructive pulmonary disease: A causal graph modeling study

**Robert W. Gregg**[1,2], **Chad M. Karoleski**[3], **Edwin K. Silverman**[4], **Frank C. Sciurba**[3], **Dawn L. DeMeo**[4], **Panayiotis V. Benos**[1] *

**1** Department of Epidemiology, University of Florida, Gainesville, Florida, United States of America, **2** Department of Biomedical Informatics, University of Pittsburgh, Pittsburgh, Pennsylvania, United States of America, **3** University of Pittsburgh Medical Center, Department of Medicine, Department of Pulmonary Allergy and Critical Care Medicine, Pittsburgh, Pennsylvania, United States of America, **4** Channing Division of Network Medicine and the Division of Pulmonary and Critical Care Medicine, Department of Medicine, Brigham and Women's Hospital, Harvard Medical School, Boston, Massachusetts, United States of America

\* pbenos@ufl.edu

## Abstract

### Background

Beyond exposure to cigarette smoking and aging, the factors that influence lung function decline to incident chronic obstructive pulmonary disease (COPD) remain unclear. Advancements have been made in categorizing COPD into emphysema and airway predominant disease subtypes; however, predicting which healthy individuals will progress to COPD is difficult because they can exhibit profoundly different disease trajectories despite similar initial risk factors. This study aimed to identify clinical, genetic, and radiological features that are directly linked—and subsequently predict—abnormal lung function.

### Methods and findings

We employed graph modeling on 2,643 COPDGene participants (aged 45 to 80 years, 51.25% female, 35.1% African Americans; enrollment 11/2007–4/2011) with smoking history but normal spirometry at study enrollment to identify variables that are directly linked to future lung function abnormalities. We developed logistic regression and random forest predictive models for distinguishing individuals who maintain lung function from those who decline. Of the 131 variables analyzed, 6 were identified as informative to future lung function abnormalities, namely forced expiratory flow in the middle range ($FEF_{25-75\%}$), average lung wall thickness in a 10 mm radius (Pi10), severe emphysema, age, sex, and height. We investigated whether these features predict individuals leaving GOLD 0 status (normal spirometry according to Global Initiative for Obstructive Lung Disease (GOLD) criteria). Linear models, trained with these features, were quite predictive (area under receiver operator characteristic curve or AUROC = 0.75). Random forest predictors performed similarly to logistic regression (AUROC = 0.7), indicating that no significant nonlinear effects were present. The results were externally validated on 150 participants from Specialized Center for Clinically Oriented Research (SCCOR) cohort (aged 45 to 80 years, 52.7% female, 4.7%

**Data Availability Statement:** COPDGene clinical, imaging, and genetic data are available through dbGAP accession number phs000179.v6.p2. The

consent for the SCCOR cohort does not permit the data to be made publicly available. The SCCOR data used here can be made available upon reasonable request by the PI of the study. Contact person: Rhonda Lincon (e-mail: lincolnra@upmc.edu).

**Funding:** This study was funded by the National Heart, Lung, Blood Institute (NHLBI) of the National Institutes of Health (NIH) with award numbers R01HL157879 and R01HL159805 (PVB), P01HL132825 (DLD), R01HL152735, P01HL114501. DLD was also supported by award number R01HG011393 from the National Human Genome Research Institute (NHGRI). RWG was support by a fellowship from the National Library of Medicine (NLM) with award number T15LM007059. The COPDGene project was supported by grants from the NHLBI (U01HL089897 and U01HL089856), by NIH contract 75N92023D00011, and by the COPD Foundation. COPDGene is supported by the COPD Foundation through contributions made to an Industry Advisory Board that has included AstraZeneca, Bayer Pharmaceuticals, Boehringer-Ingelheim, Genentech, GlaxoSmithKline, Novartis, Pfizer, and Sunovion. The funders had no role in study design, data collection and analysis, decision to publish, or preparation of the manuscript.

**Competing interests:** RWG, CMK, and PVB have no competing interests. FCS has received grant support and consulting fees from Sanofi/Regeneron, AstraZeneca, Verona Pharma, Nuvaira, Gala Therapeutics, GlaxoSmithKline, Boehringer Ingelheim. EKS has received grant support from Bayer and Northpond Laboratories. DLD has received grant support from Bayer and the Alpha-1 Foundation.

**Abbreviations:** AUROC, area under receiver operator characteristic curve; COPD, chronic obstructive pulmonary disease; CT, computed tomography; FEF, forced expiratory flow; FEV, forced expiratory volume; FVC, forced vital capacity; GOLD, Global Initiative for Obstructive Lung Disease; GWAS, Genome Wide Association Study; PEF, peak expiratory flow; SCCOR, Specialized Center for Clinically Oriented Research; SNP, single-nucleotide polymorphism.

African Americans; enrollment: 7/2007–12/2012) (AUROC = 0.89). The main limitation of longitudinal studies with 5- and 10-year follow-up is the introduction of mortality bias that disproportionately affects the more severe cases. However, our study focused on spirometrically normal individuals, who have a lower mortality rate. Another limitation is the use of strict criteria to define spirometrically normal individuals, which was unavoidable when studying factors associated with changes in normalized forced expiratory volume in 1 s ($FEV_1$%predicted) or the ratio of $FEV_1$/FVC (forced vital capacity).

## Conclusions

This study took an agnostic approach to identify which baseline measurements differentiate and predict the early stages of lung function decline in individuals with previous smoking history. Our analysis suggests that emphysema affects obstruction onset, while airway predominant pathology may play a more important role in future $FEV_1$ (%predicted) decline without obstruction, and $FEF_{25-75\%}$ may affect both.

---

## Author summary

### Why was this study done?

- Despite the number of proposed models that predict chronic obstructive pulmonary disease (COPD) progression or mortality, little research has been devoted to identifying factors affecting the onset of lung function decline.

- If we can pinpoint measurements that predict initial lung function decline, we can potentially make more informed clinical decisions by identifying at risk individuals.

### What did the researchers do and find?

- By analyzing data of 2 cohorts (COPDGene and SCCOR for discovery and validation, respectively), we found that presence of mild or worse visual computed tomography (CT) emphysema in non-obstructed individuals is a direct indicator of future obstruction, defined by forced expiratory volume in 1 s/forced vital capacity (FEV1/FVC) ratio <0.7.

- Airway wall thickness indicated future loss of FEV1%predicted with preservation of ratio owing to symmetric changes in FVC.

- Forced expiratory flow in the middle range ($FEF_{25-75\%}$) appeared to be a general direct indicator of future lung function decline.

### What do these findings mean?

- Methods like graphical models could be used to identify relevant features for prediction.

- $FEF_{25-75\%}$, average lung wall thickness in a 10 mm radius (Pi10) and attacks of wheezing/whistling in the chest are factors to consider for assessing incident lung function decline.

- While we can successfully identify measurements that predict any incident lung function decline, future studies will need to focus on predicting specific trajectories of decline.

- The main limitation of longitudinal studies with 5- and 10-year follow-up is the introduction of mortality bias that disproportionately affects the more severe cases.

## Introduction

Chronic obstructive pulmonary disease (COPD) is a lung condition that claims the lives of 3 million people each year [1], making it the fourth leading cause of death worldwide. The disease is variably characterized by alveolar destruction (emphysema), respiratory symptoms such as chronic bronchitis, and exacerbations that include episodes of increased cough, phlegm, and/or dyspnea. One approach to categorize COPD severity uses the Global Initiative for Obstructive Lung Disease (GOLD) criteria [2] which requires 2 spirometric measurements: $FEV_1$ (%predicted) (forced expiratory volume in 1 s normalized by population) and $FEV_1$/FVC ratio ($FEV_1$ over forced vital capacity) generates categories that range from normal spirometry (which we will refer to as GOLD 0) where people are considered at risk for developing COPD, to GOLD 4, indicating very severe disease. Despite these well-established categories, COPD progression is difficult to predict [3]. Some individuals maintain GOLD 0 status for years with or without symptomatic burden, some rapidly progress to severe disease with early and progressive decline in $FEV_1$ (%predicted) and $FEV_1$/FVC ratio, while others follow a different path with symmetric $FEV_1$ and FVC decline (i.e., preserved ratio impaired spirometry, PRISm). This creates a significant challenge for clinicians who need to identify individuals at risk for COPD and for translational researchers who aim to determine the mechanisms driving COPD and its progression.

Uncertainty in predicting incident COPD stems partly from disease heterogeneity [4,5]. COPD is now considered a syndrome consisting of several disease subtypes, meaning individuals with similar disease characteristics can diverge over time [6]. This heterogeneity has led to research in characterizing COPD endotypes [7,8]. Substantial progress has been made to identify biomarkers that coincide with COPD progression (including clinical, genetic, epigenetic, radiological, spirometric, and lung microbiome measurements) [9–12], but few approaches address early stages of progression. Unfortunately, including every measurement into a subtyping model obfuscates the underlying causes that differentiate individuals. Some methods, like LASSO regression, compensate through sparsity constraints, but this simply identifies variables correlated to outcomes as opposed to potential causal relationships.

To infer direct effectors and determine how they influence progression to COPD, probabilistic directed graphs (also known as "causal graphs") can be used. Given observational data, these algorithms perform conditional independence tests to distinguish true direct interactions from simple correlations [13]. The increasing volume and interoperability of multi-omics data sets enable graph-based algorithms to better identify important relationships across biological systems [14–16]. In this study, we leveraged these properties to discover which clinical, radiological, and/or genetic measurements exhibit direct relationships to abnormal lung function decline and determined if those variables can efficiently predict individuals leaving GOLD 0 status. We compared our linear graph-based predictive model with random forest modeling, which accommodates nonlinear associations. We also tested whether our causal modeling

framework will generate better predictions than random forest modeling using substantially fewer variables.

## Materials and methods

### COPDGene study population and data processing

The Genetic Epidemiology of COPD Study (COPDGene; NCT00608764) is a multicenter, longitudinal, observational study aimed at identifying COPD subtypes based on clinical measurements, chest computed tomography (CT) phenotypes, and genetic variants [17]. A total of 10,198 eligible African Americans and non-Hispanic whites with a minimum 10 pack-year smoking history across the full spectrum from normal spirometry to very severe lung abnormalities were recruited from 2008 to 2011 and followed-up every 5 years. All protocols were approved by institutional review boards and participants provided informed written consent.

Our study focused on individuals with normal spirometry (post-bronchodilator FEV1/FVC ≥0.7, FEV1 ≥80% predicted, referred to as GOLD 0) on their initial visit (**Fig 1A**, green area), resulting in 2,643 individuals (**Table 1**), 79.6% of which remained in GOLD 0 at Visit 2 (5-years) (**Figs 1B** and **S1**). Abnormal lung function is defined by change in GOLD 0 status

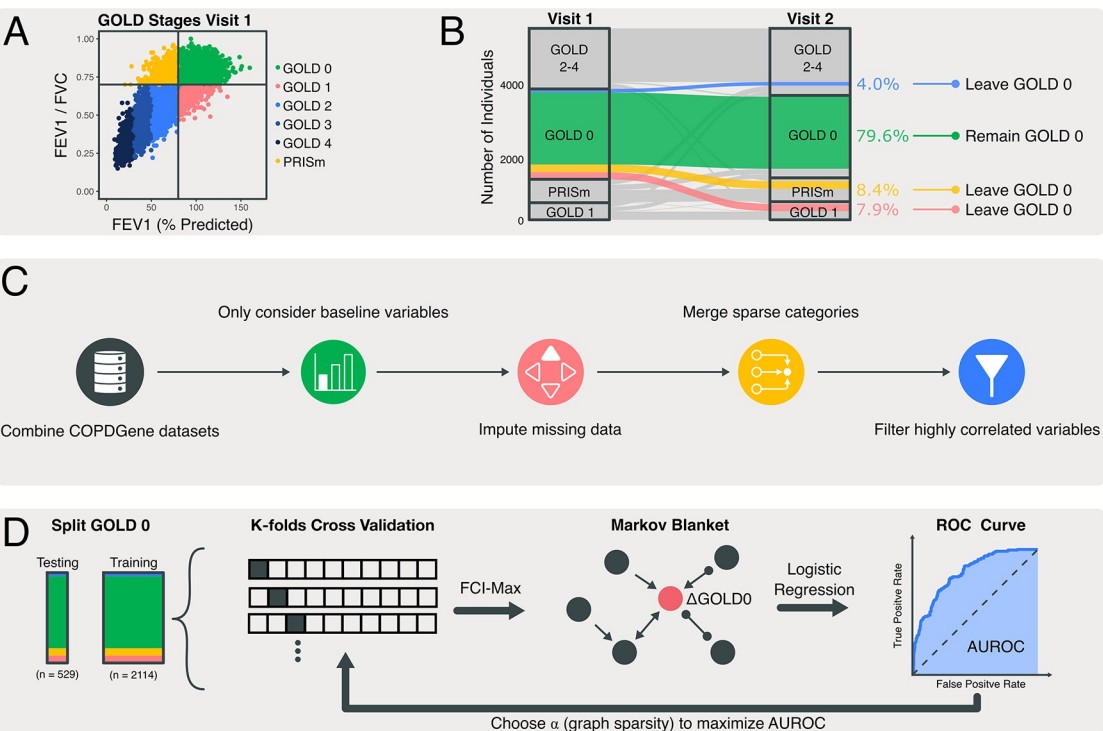

**Fig 1. Overview of the methods used to predict GOLD 0 stage progression. (A)** Shows the COPD disease axes used to categorize individuals into different GOLD stages. **(B)** Depicts the transition in GOLD stage between the first and second COPDGene visit (a 5-year difference on average); 79.6% of individuals maintain GOLD 0 status and the remaining progress to more severe disease. **(C)** Details the workflow for data processing. Combined COPDGene data sets include demographic, spirometric, radiological, genetic, and survey information. Baseline variables only include measurements from the first visit. Imputation was performed using K-nearest neighbors. Categorical variables with more than 3 levels were merged. Correlation was determined using coefficient of determination for continuous variables and Cramer's V-score for categorical variables. **(D)** Lays out the process for parameter tuning. Visit 1 data is split into a testing and training set, the latter is further divided for k-fold cross validation. For each fold and choice of α (graph sparsity), FCImax is used to determine variables causally linked to COPD progression (i.e., the Markov blanket). These variables are then used as predictors in a classification model where α is chosen by maximizing AUROC. AUROC, area under receiver operator characteristic curve; $FEV_1$, forced expiratory volume in 1 s; FVC, forced vital capacity; GOLD, Global Initiative for Obstructive Lung Disease.

**Table 1. Demographics and clinical measurements for individuals with GOLD 0 status in each data set.** Individuals were split into different cohorts stratified by GOLD 0 status. Data from COPDGene were used for Training, Testing, and Internal validation. Data from SCCOR cohort were used for External validation.

| Characteristics | Training | Testing | Internal validation | External validation |
|---|---|---|---|---|
| | (*n* = 2,114) | (*n* = 529) | (*n* = 471) | (*n* = 150) |
| Leave GOLD0 status, *n* (%) | 430 (20.3%) | 108 (20.4%) | 102 (21.7%) | 25 (16.7%) |
| Mean age, years (±SD) | 57.6 (± 8.46) | 57.9 (± 8.12) | 63.7 (± 8.18) | 64.0 (± 5.66) |
| Biological sex female, *n* (%) | 1,083 (51.2%) | 271 (51.2%) | 258 (54.8%) | 79 (52.7%) |
| Pack years, median (IQR) | 33.6 (23.3) | 34.0 (22.0) | 36.1 (22.7) | 40 (29.5) |
| Current smokers, n (%) | 1,048 (49.6%) | 243 (45.9%) | 167 (35.5%) | 63 (42.0%) |
| Race AA, n (%) | 730 (34.5%) | 197 (37.2%) | 131 (27.8%) | 7 (4.66%) |
| Mean $FEV_1$%predicted (±SD) | 97.4 (± 11.5) | 97.4 (± 11.3) | 97.5 (± 11.0) | 98.1 (±10.3) |
| Mean $FEV_1$/FVC (±SD) | 0.786 (± 0.0521) | 0.784 (± 0.0488) | 0.779 (± 0.0500) | 0.776 (±0.0427) |

*n*, count; AA, African American; COPD, chronic obstructive pulmonary disease; $FEV_1$, forced expiratory volume in 1 s; FVC, forced vital capacity; GOLD, Global Initiative for Obstructive Lung Disease; IQR, interquartile range; SCCOR, Specialized Center for Clinically Oriented Research; SD, standard deviation.

(ΔGOLD0) between baseline and the 5-year follow-up visit. ΔGOLD0 is represented by a binary variable (maintaining GOLD 0 status or not). This data division led to a class imbalance as most individuals remained in GOLD 0 (**S1B Fig**). To compensate for this imbalance, synthetic minority oversampling (SMOTE-NC) was employed [18]. This was advantageous when compared to simpler approaches like down-sampling because no data is excluded, but has its limitations when samples are too sparse or noisy [19].

A series of steps were taken to ensure data quality (**Fig 1C**). COPDGene contains 542 baseline-measured features including clinical measurements, variables derived from chest CT images, and questionnaires. We also included 81 single-nucleotide polymorphisms (SNPs) from a Genome Wide Association Study (GWAS) [20], which focused on lung function decline. This resulted in a total of 623 initial baseline variables. After filtering, 131 variables were retained (see **S2 Fig** for data filtering process) and missing values were imputed with a *k*-nearest neighbors algorithm (*k* = 5) using the Gower's distance to calculate differences between mixed data types. This imputation was performed using the *smotenc* function from the *themis* R package (v1.0.2).

Categorical variables can be problematic for probabilistic graphical algorithms because they exponentially increase the number of conditional independence tests performed. To overcome this problem, sparse categories were merged, which resulted in all categorical variables having 2 or 3 categories (affecting 13 variables).

Since GOLD 0 status is defined by 2 spirometric indices ($FEV_1$/FVC, $FEV_1$%predicted) and we did not want to bias our predictive models, we excluded FEV1, FVC, and their derivatives from our data set. This main data set is referred to as the "limited spirometry" data set. To further eliminate any effect of spirometry on the predictive models, a second data set ("no spirometry" data set) was created by excluding all other spirometry-related variables (e.g., $FEF_{25-75\%}$). This data set was used to identify associations with variables other than pulmonary function measurements.

## SCCOR study population

The Specialized Center for Clinically Oriented Research (SCCOR) study is a single-center observational cohort study of current and former smokers at the University of Pittsburgh (NCT02898272). Participants were recruited between 2007 and 2012 and followed up 2, 6, 8, and 10 years after the initial visit [21]. All procedures involved with this cohort were approved

by University of Pittsburgh Institutional Review Board. To best match the COPDGene study, initial and 6-year follow-up measurements were included in this analysis. The SCCOR cohort was used as an additional, independent validation cohort. It consisted of 150 individuals, 25 of which showed lung function abnormalities in the 6-year follow-up visit (**Table 1**).

## Causal graph algorithm

Probabilistic directed graphs (also named "causal graphs") have become popular recently due to their interpretability and ability to identify direct effects between variables even from observational data [22]. Here, we used the MGM-FCImax algorithm [23]. Both components, MGM and FCImax, have been described previously (see Sedgewick and colleagues [24,25] and Spirtes and colleagues [26]), but to summarize, MGM first determines a sparse undirected graph which contains a superset of edges present in the true directed graph. FCImax uses this network to perform independence tests to remove and orient edges. This method reduces the total conditional independence tests required and operates on mixed-data types [27].

FCImax has a parameter $\alpha$ which regulates graph sparsity (larger $\alpha$ values tend to generate graphs with more connections). Nested cross-validation was performed across 10 $\alpha$ values, logarithmically distributed between $10^{-5}$ and $10^{-1}$, and the optimal $\alpha$ value was determined by maximizing the area under the receiver operator characteristic curve (AUROC) in the predictive model. For more information, see the Supplemental Methods.

## Predictive models of future lung dysfunction

A logistic regression model was developed in R (v4.1.2) using the *glmnet* (v4.1.3) package to predict $\Delta$GOLD0 using variables from the Markov blanket identified by MGM-FCImax. In principle, a Markov blanket includes all variables containing unique information about the target variable ($\Delta$GOLD0) that cannot be obtained from any other variable or combination of variables [22,28]. To minimize overfitting, models were parameterized in a nested cross-validation setup where Markov blankets were extracted for each fold. AUROC was used to compare model performance (**Fig 1D**).

Graph algorithms coupled with logistic regression can successfully perform feature selection, make predictions, and be easily interpreted. However, this method assumes linear relationships between predictors and outcomes. To investigate the impact of nonlinear effects, we trained a separate random forest model (package *ranger* v.0.14.1) using the same data sets, training/testing splits, and 10-fold cross-validation scheme to tune associated parameters and measure performance. Assessment of the important features in the random forest models was done with Shapley values using the R *fastshap* package (v0.0.3.9). We also compared the results of our graph-based model to a regularization-based model (elastic net). This was to ensure that no information was lost due to a stricter parameter selection. For the elastic net model, we used *tidymodels* R package (v1.1.1) in conjunction with the *glmnet* package (v4.1.8) to fit the model.

## Model validation procedure

For internal validation, we applied the baseline-trained model to COPDGene individuals categorized as GOLD 0 at the 5-year follow-up visit and predicted $\Delta$GOLD0 at their 10-year visit (**Table 1**, Internal Validation). Some variables significantly changed between the 2 visits (e.g., age; **S3 Fig**). For missing certain 5-year measurements, the initial visit measurement was used, if appropriate (e.g., work status). Individuals were removed if no 10-year follow-up GOLD stage status was recorded resulting in 471 individuals. A second internal validation dataset included individuals from the 529 individuals in the original testing data set, who were not

used during training, if they met the criteria for the 5-year and 10-year visits described above (*n* = 77). The reason for this second validation data set is that some of the 471 individuals were part of the original training data set, and although their data have changed between visits, we wanted to avoid any data leakage.

As an external, independent data set we used SCCOR cohort individuals with GOLD 0 status at baseline and who returned for a 6-year follow-up (*n* = 150, **Table 1**). Because some variables in COPDGene were missing from SCCOR (notably, Pi10, the square root wall area thickness at an airway internal perimeter of 10 mm), the best fitting model was refit on COPD-Gene without Pi10. As an external validation, the refit model was assessed on the SCCOR cohort.

This study is reported as per the Transparent Reporting of a Multivariable Prediction Model for Individual Prognosis Or Diagnosis (TRIPOD) statement (S1 TRIPOD Checklist).

## Results

### Variables directly linked to 5-year ΔGOLD0 progression

We used the FCImax algorithm on the observational data from COPDGene Phase 1 cohort to learn the directed graph with optimal graph sparsity and extracted the Markov blanket for the ΔGOLD0 variable (see Methods). **Fig 2** visualizes the Markov blankets of the 2 models: the "limited spirometry" and the "no spirometry." For the "limited spirometry" model, informative variables of ΔGOLD0 included $FEF_{25-75\%}$, demographic measurements (age, height, sex), and CT scan-derived variables (visual severe emphysema and Pi10). Apart from height→$FEF_{25-75\%}$, all identified connections included the possibility of a latent confounder (**Fig 2A**). When all spirometric variables were removed from the data set ("no spirometry"

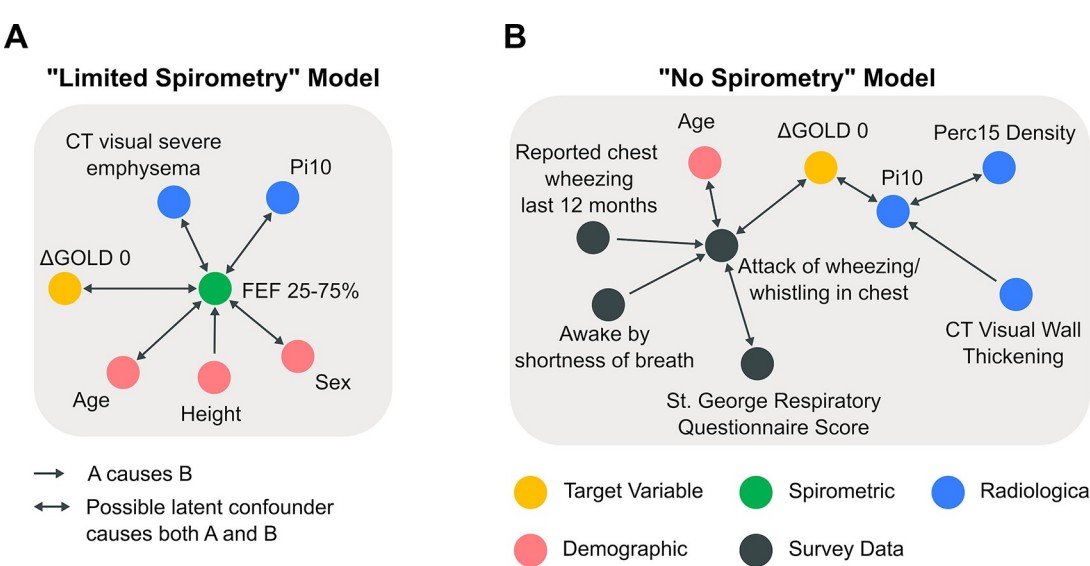

**Fig 2. Discovered Markov blanket that maximizes AUROC.** The Markov blanket encompasses every variable within the data set that can be used to infer information about our target variable: change in GOLD0 status between the first and second visit (ΔGOLD0). Each node in the graph corresponds to a measured variable in COPDGene and each edge represents a possible causal relationship that satisfies every conditional independence test performed. Arrows between variables show a direct causal link and/or an unmeasured latent confounder that causes both variables. **(A)** Depicts the optimal Markov blanket for the data set with variables related to $FEV_1$ and FVC removed. **(B)** Shows the same model, but all spirometric variables are removed. CT, computed tomography; $FEF_{25-75\%}$, forced expiratory flow in the middle range; GOLD, Global Initiative for Obstructive Lung Disease; Perc15, 15th percentile cut-off for CT lung density in Hounsfield units; Pi10, average lung wall thickness in 10 mm radius.

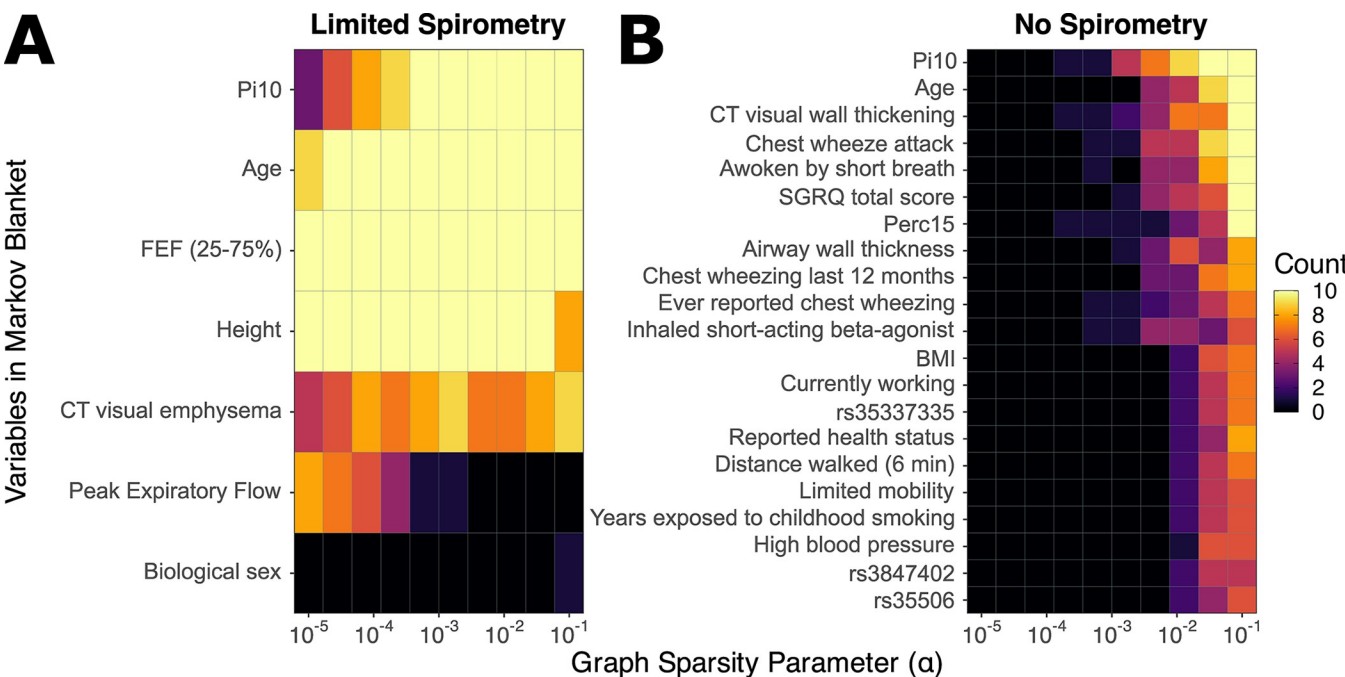

**Fig 3. Markov blanket composition across different graph sparsities.** The horizontal axis shows the log-transformed α value that controls graph sparsity (larger values lead to denser graphs), and the vertical axis shows variables that appeared in a Markov blanket. The color gradient counts the number of cross validating folds a variable appeared in the Markov blanket at that given graph sparsity. A lighter color indicates a variable appeared more frequently in the Markov blanket. **(A)** Shows the Markov blanket composition for the model with some spirometry measurements included. **(B)** Contains variables from the Markov blanket with no spirometry measurements included (showing the top 20, including ties). BMI, body mass index; CT, computed tomography; $FEF_{25-75\%}$, forced expiratory flow in the middle range; Perc15: 15th percentile cut-off for CT lung density in Hounsfield units; Pi10: average lung wall thickness in 10 mm radius; SGRQ, St George Respiratory Questionnaire.

model), informative variables included symptoms (e.g., attacks of wheezing/whistling in chest and St George Respiratory Questionnaire (SGRQ) score), age, and radiological features: visual wall thickening on CT, Pi10, and Perc15 (15th percentile cut-off for CT lung density in Hounsfield units) (**Fig 2B**).

To determine edge stability in these graphs, we measured how often edges appear across different graph sparsities and cross-folds. **Figs 3** and **S4** summarize these counts for both models. The "limited spirometry" model has fewer variables due to $FEF_{25-75\%}$ being a highly informative (and stable) predictor of ΔGOLD0 across sparsity levels (**Fig 3A**). Of note, Pi10 and peak expiratory flow (PEF) swapped places as α changed, indicating that they contain similar information about ΔGOLD0. Biological sex only appeared once at the densest graph sparsity which signified an insubstantial connection.

The "no spirometry" model was generally less stable (**S4B Fig**). Variables became connected to ΔGOLD0 at $\alpha > 10^{-4}$. Stable edge connections included Pi10, CT visual wall thickening, attacks of wheezing/whistling in chest, age, SGRQ score, and Perc15 (**Fig 3B**). In the absence of $FEF_{25-75\%}$, the "no spirometry" Markov blanket contained some SNPs, with rs35337335 (mapped to *DARS1-AS1* and *CXCR4*) being the most stable across folds.

## Linear, graph-based models predict change in GOLD 0 status

Next, we used the Markov blanket variables to build a logistic regression model for ΔGOLD0. After 10-fold cross validation, the "limited spirometry" model obtained an average AUROC of 0.75 during training, 0.75 in the test set, and 0.76 in the internal validation set, consisting of COPDGene participants in the follow-up visit (Phase 2) (**Fig 4A–4C**). This indicated that

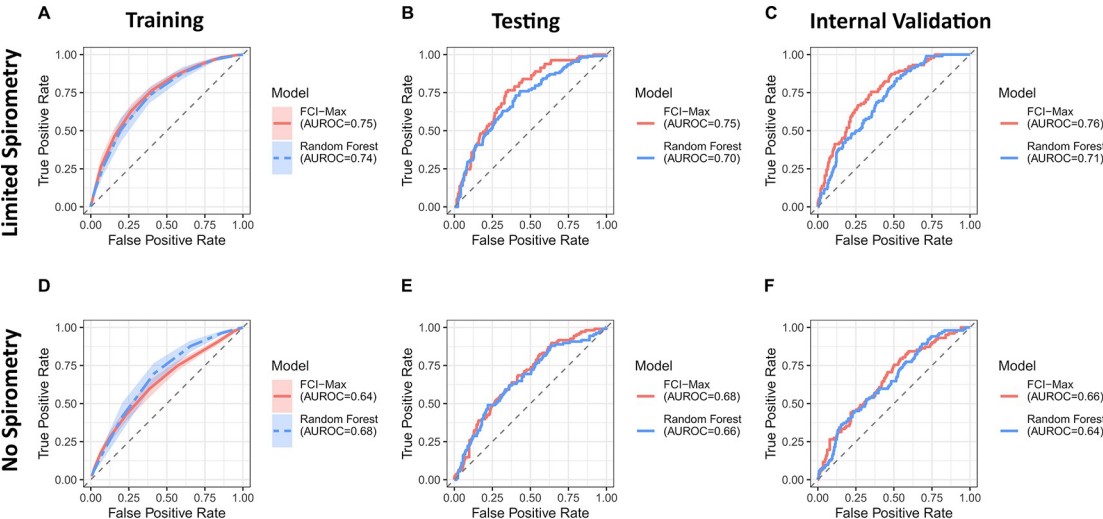

**Fig 4. Classifier performance for predicting change in GOLD 0 status.** Receiver operator characteristic curves (ROC) illustrate model performance for FCImax + Logistic Regression and random forest. The shaded areas in the training data set designate ±1 standard deviation based on 10× cross-validations. Area under the curve measurements are displayed in the legend (no significant difference). **(A–C)** Include "limited spirometry" models. **(D–F)** Include "no spirometry" models. AUROC, area under the receiver operator characteristic curve; GOLD, Global Initiative for Obstructive Lung Disease.

good predictability is not due to data overfit. The "no spirometry" model had lower performance, achieving an AUROC of 0.64, 0.68, and 0.66 for the training, testing, and internal validation sets, respectively (**Fig 4D–4F**). In order to test whether the graph-derived model contained the same predictive power as the (also linear) regularized-type predictors, we also ran an elastic net predictor and found it to perform similarly (**S5 Fig**).

Our external validation cohort (SCCOR) did not include Pi10, which is part of the "limited spirometry" model. To assess whether the COPDGene-trained model generalized to external cohorts, the variables from the "limited spirometry" model without Pi10 were used to retrain the linear model on the COPDGene training set. Applied to the SCCOR data set, the re-trained model had a small reduction in AUROC for the COPDGene testing data set (from 0.75 to 0.73) but resulted in an AUROC of 0.89 in the SCCOR data set (**Fig 5**). This indicated that our model's predictions are robust and generalize beyond COPDGene.

In an attempt to explain the higher accuracy of our model in the SCCOR (validation) cohort, we examined the separation of the individuals in the 2 categories (leave/stay GOLD 0) with respect to the strongest predictor, $FEF_{25-75\%}$. **S6 Fig** shows those that leave GOLD 0 had generally lower $FEF_{25-75\%}$ and less variance in SCCOR than those in COPDGene. Furthermore, SCCOR had a higher percentage of people staying in GOLD 0 (83.3% versus 79.7% in COPDGene).

The training and internal validation data sets had some dependency because the same individuals participate in both (although at different time points). Therefore, we formed a completely unbiased COPDGene validation cohort by using the 77 GOLD 0 individuals from the testing data set that remained GOLD 0 at the 5-year visit. These 77 individuals were not used in training. Performance on the original validation data set ($n = 471$) and the more stringent validation data set ($n = 77$) showed no substantial difference (**S7 Fig**). The "limited spirometry" graph-based model performed slightly worse (AUROC = 0.71 versus 0.76) and the "no spirometry" model slightly better (AUROC = 0.66 versus 0.68). Comparable results were seen with the random forest models.

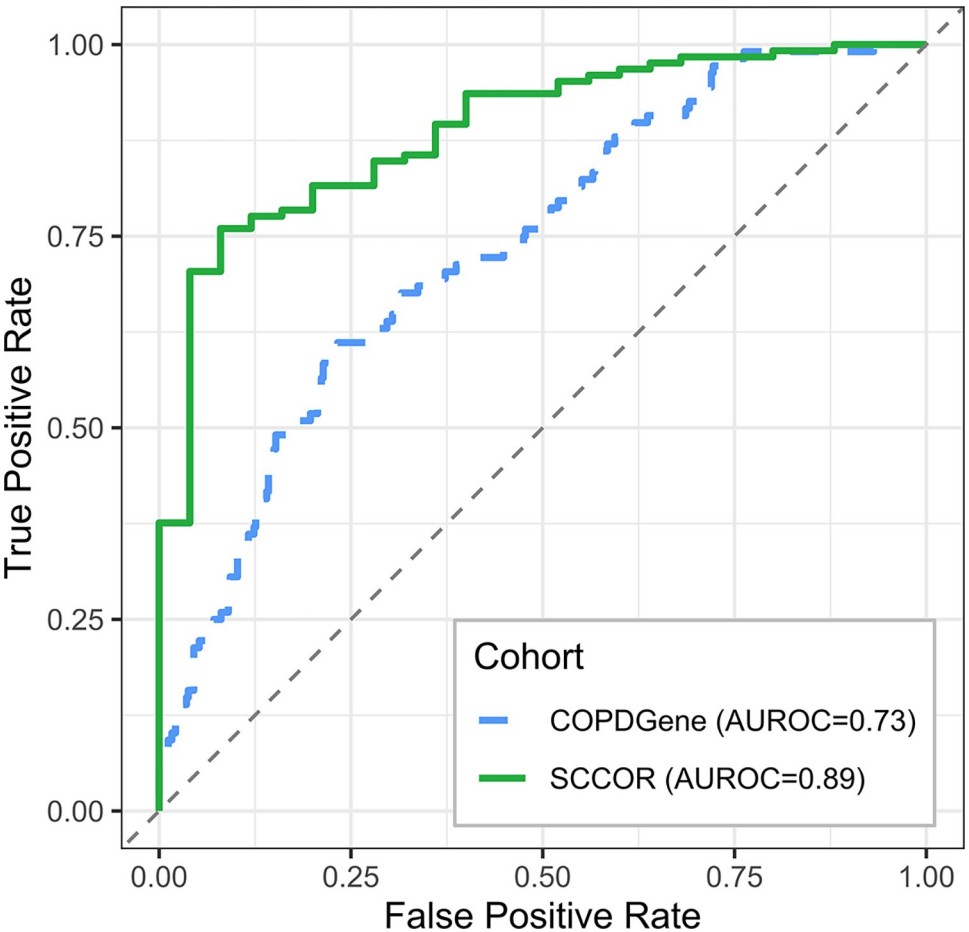

**Fig 5. Classifier performance for predicting change in GOLD 0 status in the SCCOR cohort.** Plotted are the ROC curves for COPDGene test data set and the external validation SCCOR cohort. The final model predicts change in GOLD 0 status without Pi10 as a predictor in the limited spirometry model. Each legend entry provides the area under the receiver operator curve (AUROC). AUROC, area under the receiver operator characteristic curve; SCCOR, Specialized Center for Clinically Oriented Research.

## FEF$_{25-75\%}$ as a general predictor of future lung abnormalities

To assess the significance of each of the 6 predictors identified by FCImax, we calculated a log-odds risk using measurements obtained during clinical assessment of COPD diagnosis. We found that FEF$_{25-75\%}$ had the most significant coefficient in predicting $\Delta$GOLD0, while age and medium levels of visual CT emphysema did not have significant coefficients (**Table 2**). Ideally, this model could be used as a simple estimate to identify individuals with higher odds of progressing into COPD.

In the previous analyses, we grouped together $\Delta$GOLD0 individuals with reduction in FEV$_1$%predicted only (PRISm) and those with FEV$_1$/FVC reduction (COPD, GOLD1-4). To further investigate whether some of the identified variables were specific to one of the 2 trajectories, we retrained the linear models separately in each group and examined their coefficients (**Table 2**). We found Pi10 to be not significant for the COPD-specific model but remained significant for the individuals progressing to PRISm. Furthermore, we saw that having mild or worse visual CT emphysema was more important in the COPD-specific model than the PRISm-specific model.

**Table 2. Logistic regression "limited spirometry" models.** All models trained on COPDGene data. All data model focuses on ΔGOLD0, regardless of whether the individual loses $FEV_1/FVC$ or $FEV_1$%predicted. COPD-specific model focuses on those individuals that see reduction in $FEV_1/FVC$ (<0.70). PRISm-specific model focuses on with reduction in $FEV_1$%predicted (<0.80). SE: standard error.

| Term | "Limited Spirometry" all data model (AUROC = 0.75) | | | COPD-specific model (AUROC = 0.76) | | | PRISm-specific model (AUROC = 0.66) | | |
|---|---|---|---|---|---|---|---|---|---|
| | Coefficient estimate | SE | *P*-Value | Coefficient estimate | SE | *P*-Value | Coefficient estimate | SE | *P*-Value |
| (Intercept) | −0.40 | 0.09 | $4.68 \times 10^{-6}$ | −0.53 | 0.18 | 0.0039 | −0.39 | 0.18 | 0.029 |
| $FEF_{25-75\%}$ | −1.23 | 0.14 | $< 2 \times 10^{-16}$ | −1.54 | 0.19 | $< 2 \times 10^{-16}$ | −0.99 | 0.17 | $2.46 \times 10^{-9}$ |
| Age | −0.05 | 0.10 | 0.60 | −0.09 | 0.11 | 0.41 | −0.17 | 0.12 | 0.33 |
| Height (cm) | 0.47 | 0.13 | 0.00018 | 0.58 | 0.15 | $6.52 \times 10^{-5}$ | 0.27 | 0.15 | 0.068 |
| Pi10 | 0.30 | 0.09 | 0.00074 | −0.0036 | 0.10 | 0.97 | 0.51 | 0.11 | $3.12 \times 10^{-6}$ |
| Sex (female) | −0.37 | 0.12 | 0.0017 | −1.12 | 0.26 | $2.03 \times 10^{-5}$ | −1.31 | 0.28 | $1.96 \times 10^{-6}$ |
| Visual CT Emphysema | | | | | | | | | |
| - Trace | −0.06 | 0.09 | 0.46 | −0.61 | 0.29 | $4.00 \times 10^{-6}$ | −0.79 | 0.29 | 0.007 |
| - Mild or worse | 0.36 | 0.08 | $7.62 \times 10^{-6}$ | 1.31 | 0.21 | $6.65 \times 10^{-10}$ | −0.14 | 0.26 | 0.59 |

AUROC, area under the receiver operator characteristic curve; CT, computed tomography; $FEF_{25-75\%}$, forced expiratory flow in the middle range; $FEV_1$, forced expiratory volume in 1 s; FVC, forced vital capacity; Pi10, average lung wall thickness in 10 mm radius; PRISm, preserved ratio impaired spirometry (designates change in $FEV_1$%predicted but preserved $FEV_1/FVC$ ratio); SE, standard error.

Changing status from GOLD 0 to GOLD 1–4 means reduction in $FEV_1/FVC$ ratio. Thus, one concern for the use of the $FEF_{25-75\%}$ in the "limited spirometry" model is a potential "information leakage" due to its correlation to the $FEV_1/FVC$ ratio ($R^2$ = 0.717 in our training data set). This may lead to easy predictions of transition for those individuals in the $FEV_1/FVC$ threshold between GOLD0 and GOLD1-4 (all GOLD 1–4 have $FEV_1/FVC$<0.7). To test whether our "limited spirometry" model was indirectly influenced by $FEV_1/FVC$, we plotted the individuals that remain GOLD0 or develop spirometric abnormalities in the testing data set and how well the limited spirometry model predicted them. As expected, people that left GOLD0 tended to concentrate more around the 2 spirometric thresholds, but there was a good number that were more distant (**S8A Fig**). Furthermore, of those that left GOLD0, we note that the "limited spirometry" model predicted quite well even those that were more distant to the borders (**S8B Fig**).

## Consideration of nonlinear associations does not improve ΔGOLD0 prediction

To assess whether consideration of nonlinear associations offers additional information in predicting ΔGOLD0, random forest models were fit using the same 10-fold cross validation data sets, giving similar AUROCs on the training data set (0.74 and 0.68 for the "limited" and "no spirometry" data set, respectively; **Fig 4A** and **4D**). For the COPDGene testing and validation data sets, the random forest AUROCs remained smaller than the graph-based predictions (**Fig 4**). This is despite the graph-based model using only 6 predictors, while random forest used all 131 variables (**Table 2**). These results suggested lack of strong nonlinear effects in our data set.

The top 15 most important variables affecting random forest performance in the "limited spirometry" model are presented in **Fig 6A** ("no spirometry" model in **S9A Fig**). $FEF_{25-75\%}$ and Pi10 appeared as the most important (as in the graph-based model), followed by SGRQ score. Additionally, LAA950 (percentage of low attenuation areas less than −950 Hounsfield units) appeared as one of the top features across all random forest models, but rarely in the causal graph-based models.

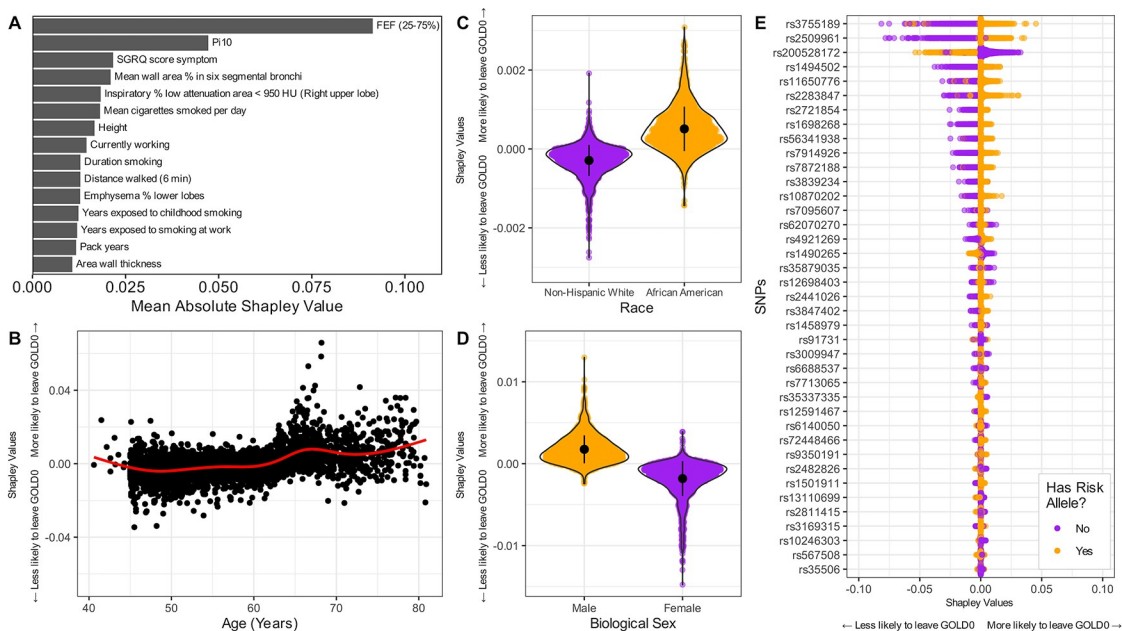

**Fig 6. Exploring variables importance to the random forest model ("limited spirometry" model). (A)** Variables are ordered by importance using mean absolute Shapley values. **(B–D)** The distribution of Shapley values across measured demographics (age, biological sex, and race). Positive Shapley values on vertical axes indicate the random forest model was more likely to predict that individual to leave the GOLD 0 status (and vice versa). In (**B**) we do not display 8 (out of 2,114) individuals that were <45 years old at baseline, since they did not match the inclusion criteria of COPDGene. **(E)** SNP contributions to the random forest model prediction. Colors differentiate individuals with and without a given SNP. CT, computed tomography; FEF$_{25-75\%}$, forced expiratory flow in the middle range; GOLD, Global Initiative for Obstructive Lung Disease; Pi10, average lung wall thickness in 10 mm radius; SGRQ, St George Respiratory Questionnaire; SNP, single-nucleotide polymorphism.

### Contribution of demographics and SNPs to the prediction of ΔGOLD0 in the random forest model

By comparing Shapley values across different demographics, we found differences for age, self-reported race, and biological sex (**Fig 6B–6D**). The random forest model assigned higher importance to older individuals in predicting ΔGOLD0, especially for those of 65 to 70 years. This was observed in the "limited spirometry" data set as well (compare **Fig 6B** with **S9B Fig**). Race and biological sex played a smaller role than age for this model. Still, being African American or male positively contributed to higher ΔGOLD0 prediction score.

Shapley values for SNPs often exhibited a visual distinction between individuals with and without a given risk allele (**Fig 6E**). One SNP with the largest absolute Shapley value was rs2509961 (*AHNAK* gene) which was one of the 2 SNPs appearing in some of the Markov blankets of the "no spirometry" model (**S4B Fig**). Finally, SNPs did not exhibit strong predictive capabilities across our models (**S9E Fig**). However, recent research suggests that polygenic risk scores (a weighted score of all SNPs adjusted for demographics and ancestry) can outperform predictions from individual SNPs [29]. To further investigate these effects, SNPs selected by the graphical model (*n* = 39) were regressed onto ΔGOLD0, but they showed low predictive power (AUROC 0.51, **S10 Fig**). The polygenic risk score from [29] showed only slightly better performance (AUROC 0.55).

### Discussion

COPD is a multifaceted disease driven by genetic, epigenetic, and environmental factors that coalesce into trajectories that are difficult to predict. Previous studies developed models to

predict COPD progression, as defined by change in FEV1. For example, Wang and colleagues [30] built a logistic regression model to predict rapid FEV1 decline after 5 years. Also, Boueiz and colleagues [31] built both linear regression and random forest models to predict change in FEV1 between 5-year follow-up visits. They found that predicting FEV1 decline is challenging even with more advanced machine learning methods. Both studies utilized a larger number of predictors than us (22 and 46, respectively).

In this study, we focused on determining potential direct effectors of development of lung function abnormalities by looking simultaneously on both spirometric measures: $FEV_1$ (%predicted) and $FEV_1/FVC$. Lung function abnormality was determined through spirometric decline in non-GOLD 0 status (binary outcome), but our methods can be applied to other definitions of disease development. With the identified potential cause–effect relationships, we predicted change in GOLD 0 status between baseline and 5-year follow-up visits.

The most predictive non-spirometric variable we identified was a quantitative imaging assessment of airway wall thickening (Pi10). Previous studies have proposed Pi10 as an early diagnostic marker of COPD [32,33]. In our models, Pi10 was found to be more important for entering PRISm following $FEV_1$ decline (see, also, [34,35]), than to onset of obstruction. Large airway wall thickening is the result of infiltrating T-cells, macrophages, and neutrophils releasing pro-inflammatory cytokines causing cell hyperplasia, mucous gland hypertrophy, thicker smooth muscle layers, and fibrosis [36,37]. Unmeasured inflammation may be the latent confounder between Pi10 and ΔGOLD0 that our model predicted. Our results indicated that airway disease predisposes people to reduction in $FEV_1$%predicted only (i.e., moving to PRISm). By contrast, we found that mild or worse emphysema is significantly affecting onset of obstruction (GOLD 1–4), but not entering the PRISm category.

Small airways maintain aberrant inflammation levels and are often associated with incident COPD [38]. This association was identified in our "limited spirometry" models, which established $FEF_{25-75\%}$ as a strong predictor for ΔGOLD0 [39]. However, whether $FEF_{25-75\%}$ is reflective of small airway disease remains controversial [40–42]. Researchers have proposed 2 COPD subtyping modalities: airway and emphysema predominant disease trajectories [43,44]. Our models supported this idea by identifying variables that were directly associated with airway predominant (Pi10) and emphysema predominant (CT visual emphysema severity) subtypes, while we found $FEF_{25-75\%}$ to be important effector of both. Regardless, variables associated with airway predominant disease were more prevalent and predictive of ΔGOLD0.

LAA950 was identified as a strong predictor in the random forest "limited spirometry" model, but our graph-based model suggested that it is not directly linked to ΔGOLD0. Instead, other radiographic variables (Pi10 and visual CT emphysema) contained potential cause–effect relationships that require further investigation. Additionally, we found that graph-based predictions in simple logistic regression models compete with random forest modeling despite stringent variable selection in the former. Overall, causal graph algorithms did not always improve model predictive performance, but they excelled at identifying the most pertinent and predictive variables and provided interpretable results to generate hypotheses.

Being able to predict who is going to progress is of high value for people at risk of developing COPD. However, our model's clinical utility is restricted by the limited racial diversity in our training cohorts and the fact that all participants were older with extensive history of smoking (>10 pack-years), which makes them high risk. This impacts current clinical utility until the model is evaluated in diverse cohorts and cohorts with lower and no smoking histories. Another limitation is that Pi10, which was an important contributor to our predictive models, can only be obtained through CT scan. This limits the models' applicability in resource limited settings. Furthermore, although using GOLD stage to define COPD progression is advantageous because it creates a binary outcome to predict, it has some limitations

due to strict cut-offs that define progression. Individuals can experience loss in lung function but remain in the GOLD 0 status. Some studies [45] circumvent this problem by not discretizing participants into stages and instead predict spirometric measurements as continuous variables. However, this can be challenging because the model needs to predict both the magnitude and direction of spirometric change in both axes ($FEV_1$%predicted and $FEV_1$/FVC). Ideally, multiple criteria (symptom development, changes in lung parenchyma, spirometry, etc.) need to be combined to describe COPD progression, similar to recommendations made for COPD diagnosis [46].

Another limitation is dropout rate between visits. In COPDGene, 47% of participants have an unknown GOLD stage status by their second visit. This may introduce a bias as models are not trained on individuals with rapid COPD progression leading to death within 5 years. However, normal spirometry individuals (GOLD 0) have lower mortality than the other categories, so we expect that the dropouts did not have a major impact in our study.

Our internal and external model validations suggest that our model predictions can generalize beyond the original training data set. Internal model predictions resulted in similar AUROC values for the 10-year follow-up and they increased for the external SCCOR data set. This improvement in model performance may be partially explained by the lower percentage of individuals that left GOLD 0 status (20.3% versus 16.7%) and the lower overall $FEF_{25-75\%}$ values observed in those individuals. We would also expect model predictions in the SCCOR cohort to increase with the addition of Pi10 as a predictor.

Knowing which variables provide direct information (i.e., beyond correlates) to COPD onset helps explain the heterogeneity observed in disease progression. This paper showed the limits of predictive models that use only clinical and standard radiomics features. Investigation beyond clinical measurements is needed, but knowing which mechanisms relate to incident COPD is valuable for identifying individuals most at risk. Overall, this study found a small subset of variables that may identify individuals at risk for progression to COPD, which could help clinicians develop new disease management protocols.

## Supporting information

**S1 TRIPOD Checklist. The TRIPOD Checklist for development and validation of prediction models.**
(DOCX)

**S1 Fig. GOLD Stage transitions across all time points.** (A) The progression of GOLD stage is visualized for baseline (visit 1), 5-year follow-up (visit 2), and 10-year follow up (visit 3). Individuals with a missing GOLD stage status were assigned to the "Unknown" category. A total of 2,643 GOLD 0 individuals had known GOLD Stage statuses between the first and second visit while 471 GOLD 0 individuals had known GOLD Stage statuses between the second and third visit. (B) Shows the breakdown of these individuals as they transition between visits. A Chi-squared test revealed no significant difference between visit transitions ($p = 0.21$). GOLD, Global Initiative for Obstructive Lung Disease.
(PDF)

**S2 Fig. Procedure to filter baseline variables.** From the 623 baseline variables, variables were removed for having information about future visits, being duplicate/constant/uninformative, having too many missing values, having categories that were too small, or for being too correlated.
(PDF)

**S3 Fig. Age distribution across data splits.** For each data division, the distribution of age is plotted along with the mean (black circle) and standard deviation (vertical line). Mann–Whitney–Wilcoxon tests were performed for each pairwise comparison. Age was significantly different in the validation data set for each comparison. A Kruskal–Wallis test was also performed and showed global significance across data splits.
(PDF)

**S4 Fig. Markov blanket variable counts.** Each edge in the graph is labeled with the number of times it appeared in a Markov blanket across each fold and graph sparsity iteration (10 folds and 10 sparsities). (A) Shows every Markov blanket variable when only FEV1 and FVC variables were removed. (B) Shows only first neighbors to with no spirometry variables. CT, computed tomography; Pi10, average lung wall thickness in 10 mm radius; $FEF_{25-75\%}$, forced expiratory flow in the middle range; PEF, peak expiratory flow.
(PDF)

**S5 Fig. Classifier performance for predicting change in GOLD 0 status.** Receiver operator curves illustrate model performance for FCI-Max + Logistic Regression and random forest. Area under the curve measurements are displayed in the legend. (A–C) Include "limited spirometry" models. (D–F) Includes "no spirometry" models. AUROC, area under the receiver operator characteristic curve.
(PDF)

**S6 Fig. Boxplots indicating the separation with respect to the $FEF_{25-75\%}$ of the 2 classes (leave, stay in GOLD 0).** (A) COPDGene cohort, (B) SCCOR cohort. SCCOR has lower median and variability of $FEF_{25-75\%}$ and higher separation of those individuals that leave GOLD 0 (i.e., develop lung abnormalities). GOLD, Global Initiative for Obstructive Lung Disease; $FEF_{25-75\%}$, forced expiratory flow in the middle range.
(PDF)

**S7 Fig. Comparing ΔGOLD0 predictions for different validation data sets. (A, B)** The 2 model prediction results with all 471 individuals who were GOLD 0 status at the 5-year follow-up and had a known GOLD stage status at the 10-year follow-up. This is the internal validation data set. **(C, D)** The same results but on a subset of 77 five-year follow-up GOLD 0 status individuals who were part of the initial testing data set. These individuals were not used at all during training. AUROC, area under the receiver operator characteristic curve.
(PDF)

**S8 Fig.** (A) GOLD 0 COPDGene participants that develop spirometric abnormalities at 5-year follow-up visit ("Leave GOLD 0"; red dots) and those that remain GOLD 0 (gray dots). (B) Correct prediction of those "Leaving GOLD 0" by the limited spirometry model (blue dots) and misclassified individuals (orange dots). GOLD, Global Initiative for Obstructive Lung Disease; FEV1, forced expiratory volume in 1 s; FVC, forced vital capacity.
(PDF)

**S9 Fig. Exploring variable important within the random forest model ("no spirometry" model). (A)** Variables are ordered by importance using mean absolute Shapley values. **(B–D)** Show the distribution of Shapley values across measured demographics (age, biological sex, and race). Positive Shapley values on vertical axes indicate the random forest model was more likely to predict that individual to leave the GOLD 0 status (and vice versa). In (B) we do not display 4 (out of 529) individuals that were <45 years old at baseline, since they did not match the inclusion criteria of COPDGene. **(E)** Describes the contributions SNPs had to the random forest model prediction. Colors differentiate individuals with and without a given SNP. Pi10,

average lung wall thickness in 10 mm radius; HU, Hounsfield units; SGRQ, St George Respiratory Questionnaire.
(PDF)

**S10 Fig. Polygenic risk scores (PRS) outperform SNPs in classifying ΔGOLD0.** Plotted are the averaged cross fold validated ROC plots. Each legend entry provides the area under the receiver operator curve (AUROC). The dashed gray line represents an uninformative model. AUROC, area under the receiver operator characteristic curve.
(PDF)

**S1 Materials. Supplemental information for some of the methods used in the paper.** Details are provided for the nested cross-validation scheme and the elastic net regression method.
(DOCX)

## Acknowledgments

The content is solely the responsibility of the authors and does not necessarily represent official views of NHLBI or NIH.

## Author Contributions

**Conceptualization:** Robert W. Gregg, Panayiotis V. Benos.

**Data curation:** Chad M. Karoleski.

**Formal analysis:** Robert W. Gregg.

**Funding acquisition:** Edwin K. Silverman, Frank C. Sciurba, Dawn L. DeMeo, Panayiotis V. Benos.

**Investigation:** Edwin K. Silverman, Dawn L. DeMeo.

**Supervision:** Panayiotis V. Benos.

**Validation:** Robert W. Gregg, Chad M. Karoleski, Frank C. Sciurba, Panayiotis V. Benos.

**Writing – original draft:** Robert W. Gregg, Panayiotis V. Benos.

**Writing – review & editing:** Robert W. Gregg, Chad M. Karoleski, Edwin K. Silverman, Frank C. Sciurba, Dawn L. DeMeo, Panayiotis V. Benos.

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
