## [Editor Report · Decision Letter 0]

7 Feb 2024

Dear Dr Benos, 

Thank you for submitting your manuscript entitled "Causal Graph Modeling Identifies Six Factors Linked to Incident COPD" for consideration by PLOS Medicine.

Your manuscript has now been evaluated by the PLOS Medicine editorial staff and I am writing to let you know that we would like to send your submission out for external peer review.

Please re-submit your manuscript within two working days, i.e. by Feb 09 2024 11:59PM.

Kind regards,

Alexandra Schaefer, PhD

Associate Editor

PLOS Medicine

aschaefer@plos.org

---

## [Decision Letter · Decision Letter 1]

14 May 2024

Dear Dr. Benos,

Thank you very much for submitting your manuscript "Causal Graph Modeling Identifies Six Factors Linked to Incident COPD" (PMEDICINE-D-24-00407R1) for consideration at PLOS Medicine. 

We have considered your appeal request and I am writing to let you know that we are willing to reconsider the manuscript. Please note that the manuscript will undergo further external review before a decision is made regarding publication.

As you know, your paper was evaluated by an associate editor and discussed among all the editors here. It was also discussed with an academic editor with relevant expertise, and sent to independent reviewers, including a statistical reviewer. The reviews are appended at the bottom of this email and any accompanying reviewer attachments can be seen via the link below:

[LINK]

In light of these reviews, I am afraid that we will not be able to accept the manuscript for publication in the journal in its current form, but we would like to consider a revised version that addresses the reviewers' and editors' comments. Obviously we cannot make any decision about publication until we have seen the revised manuscript and your response, and we plan to seek re-review by one or more of the reviewers. 

In revising the manuscript for further consideration, your revisions should address the specific points made by each reviewer and the editors. Please also check the guidelines for revised papers at http://journals.plos.org/plosmedicine/s/revising-your-manuscript for any that apply to your paper. In your rebuttal letter you should indicate your response to the reviewers' and editors' comments, the changes you have made in the manuscript, and include either an excerpt of the revised text or the location (eg: page and line number) where each change can be found. Please submit a clean version of the paper as the main article file; a version with changes marked should be uploaded as a marked up manuscript. You may submit some of the files provided in your appeal request, if appropriate. 

Please use the following link to submit the revised manuscript: https://www.editorialmanager.com/pmedicine/

We expect to receive your revised manuscript by Jun 04 2024. However, if this deadline is not feasible, please contact me by email, and we can discuss a suitable alternative.

Don't hesitate to contact me directly with any questions (aschaefer@plos.org). If you reply directly to this message, please be sure to 'Reply All' so your message comes directly to my inbox.

We look forward to receiving your revised manuscript. 

Sincerely,

Alexandra Schaefer, PhD

PLOS Medicine

plosmedicine.org

***Please note: not all will apply to your paper, but please check each item carefully

FINANCIAL DISCLOSURE

The funding statement should include: specific grant numbers, initials of authors who received each award, URLs to sponsors’ websites. Also, please state whether any sponsors or funders (other than the named authors) played any role in study design, data collection and analysis, the decision to publish, or preparation of the manuscript. If they had no role in the research, include this sentence: “The funders had no role in study design, data collection and analysis, decision to publish, or preparation of the manuscript.”

COMPETING INTEREST

All authors must declare their relevant competing interests per the PLOS policy, which can be seen here: https://journals.plos.org/plosmedicine/s/competing-interests

For authors with ties to industry, please indicate whether any of the interests has a financial stake in the results of the current study.

DATA AVAILABILITY STATEMENT

For each data source used in your study:

FORMATTING 

Abstract: Please structure your abstract using the PLOS Medicine headings (Background, Methods and Findings, Conclusions). Please combine the Methods and Findings sections into one section, “Methods and findings”. 

At this stage, we ask that you include a short, non-technical Author Summary of your research to make findings accessible to a wide audience that includes both scientists and non-scientists. The Author Summary should immediately follow the Abstract in your revised manuscript. This text is subject to editorial change and should be distinct from the scientific abstract. Ideally each sub-heading should contain 2-3 single sentence, concise bullet points containing the most salient points from your study. In the final bullet point of ‘What Do These Findings Mean?’, please include the main limitations of the study in non-technical language. Please see our author guidelines for more information: https://journals.plos.org/plosmedicine/s/revising-your-manuscript#loc-author-summary. 

Please express the main results with 95% CIs as well as p values. When reporting p values please report as p<0.001 and where higher as the exact p value p=0.002, for example. Throughout, suggest reporting statistical information as follows to improve clarity for the reader “22% (95% CI [13%,28%]; p</=)”. Please be sure to define all numerical values at first use. 

Please include page numbers and line numbers in the manuscript file. Use continuous line numbers (do not restart the numbering on each page). 

Please cite the reference numbers in square brackets. Citations should precede punctuation. 

FIGURES AND TABLES 

Please provide titles and legends for all figures and tables (including those in Supporting Information files). 

Please define all abbreviations used in each figure/table (including those in Supporting Information files). 

Please consider avoiding the use of red and green in order to make your figure more accessible to those with color blindness. 

SUPPLEMENTARY MATERIAL 

Please amend THE supplementary material according to the relevant comments. 

Please cite your Supporting Information as outlined here: https://journals.plos.org/plosmedicine/s/supporting-information

REFERENCES 

PLOS uses the numbered citation (citation-sequence) method and first six authors, et al. 

Please ensure that journal name abbreviations match those found in the National Center for Biotechnology Information (NCBI) databases (http://www.ncbi.nlm.nih.gov/nlmcatalog/journals), and are appropriately formatted and capitalised. 

Where website addresses are cited, please include the complete URL and specify the date of access (e.g. [accessed: 12/06/2023]). 

Please also see https://journals.plos.org/plosmedicine/s/submission-guidelines#loc-references for further details on reference formatting. 

STUDY-SPECIFIC REQUESTS

The following text is derived from Geoffrey P Garnett, Simon Cousens, Timothy B Hallett, Richard Steketee, Neff Walker. Mathematical models in the evaluation of health programmes. (2011) Lancet DOI:10.1016/S0140-6736(10)61505-X: 

Please provide a diagram that shows the model structure, including how the disease natural history is represented, the process and determinants of disease acquisition, and how the putative intervention could affect the system. 

Please provide a complete list of model parameters, including clear and precise descriptions of the meaning of each parameter, together with the values or ranges for each, with justification or the primary source cited, and important caveats about the use of these values noted. 

Please provide a clear statement about how the model was fitted to the data, including goodness-of-fit measure, the numerical algorithm used, which parameter varied, constraints imposed on parameter values, and starting conditions. 

For uncertainty analyses, please state the sources of uncertainties quantified and not quantified [can include parameter, data, and model structure]. 

Please provide sensitivity analyses to identify which parameter values are most important in the model. Uncertainty estimates seek to derive a range of credible results on the basis of an exploration of the range of reasonable parameter values. The choice of method should be presented and justified. 

Please discuss the scientific rationale for this choice of model structure and identify points where this choice could influence conclusions drawn. Please also describe the strength of the scientific basis underlying the key model assumptions. 

For studies that develop a prediction model or evaluate its performance, please ensure that the study is reported according to the TRIPOD statement (available from https://www.equator-network.org/reporting-guidelines/tripod-statement) and include the completed checklist as Supporting Information. Please add the following statement, or similar, to the Methods: "This study is reported as per the Transparent Reporting of a Multivariable Prediction Model for Individual Prognosis Or Diagnosis (TRIPOD) statement (S1 Checklist)." When completing the checklist, please use section and paragraph numbers, rather than page numbers. 

Comments from the reviewers:

Reviewer #1: "Causal Graph Modeling Identifies Six Factors Linked to Incident COPD" describes the development of a linear (graph-based) logistic regression (LR) model using six input variables, for predicting the progression of chronic obstructive pulmonary disease (COPD) over five/ten years. In particular, the graph-based causal modelling to identify six key variables was found to perform similarly to random forest (RF) models utilizing all 131 variables on both a COPDGene validation dataset and an external SCCOR dataset, which was claimed to indicate the lack of significant non-linear associations, and also the viability of implementing a simple regression model for COPD progression. The reduction in number of required variables would allow the model to be more accessible, and possibly generalizable (due to reduced chance of overfitting).

While the determination of causality amongst variables in COPD progression is well-motivated, some issues might be considered:

1. In the Abstract and Introduction, metrics such as "FEV1" and "FVC" are introduced without explaining their significance. Such metrics & abbreviations might be explained at first occurence.

2. In the Introduction, it is claimed that "Some methods, like LASSO regression, compensate through sparsity constraints, but this simply identifies variables correlated to outcomes as opposed to potential causal relationships". However, it is not empirically established that variable correlation is inferior to causal determination, in the study that follows. Since the shortcoming of LASSO regression/correlation is claimed, such methods might be included in the experiments (together with RF).

3. Related to the above, on the main claim that logistic regression models perform similarly to RF (and thus no significant non-linear associations for the COPD progression task), the most relevant comparison between LR with 6 input variables and RF with 6 input variables, appears not to have been attempted (the comparison was against RF with 131 input variables). This might be included to confirm the non-inferiority of LR given identical inputs.

4. In the COPDGene Study Population subsection, there appears a significant proportion of dropouts (i.e. transition to "Unknown" status, in Supplemental Figure S1) over 5 and 10 year visits. Potential bias in the dropout population (possibly due to COPD progression between visits likely resulting in increased risk of death/non-followup) might be discussed.

5. In the COPDGene Study Population subsection, it is stated that missing values were imputed with a k-nearest neighbours (k=5) algorithm. It might be clarified as to how imputation was performed for categorical data, with possible ties (e.g. 2-2-1 or 1-1-1-1-1, with a tie in the most common category)

6. The SCCOR dataset used for external validation appears to be of a similar (American) demographic, as the COPDGene development dataset. It might be considered to include external validation on additional populations/demographics if possible, to further support the generalizability of the model and results.

7. In the Causal Graph Algorithm subsection, it is stated that nested cross-validation was performed across ten a (graph sparsity parameter) values, maximizing AUROC. The nested cross-validation procedure (and data on which the AUROC was computed on) might be described in greater detail.

8. It is then stated that AUROC was also maximized for logistic regression models using variables from a covering Markov blanked. Again, the nested cross-validation procedure might be described in greater detail. Moreover, would it have been more appropriate to jointly optimize the logistic regression model(s) together with the graph sparsity parameter a?

9. It is stated that 10-fold cross-validation was used to train the RF model(s). Was this equivalent to the "nested cross-validation" previously described, and if not, what was the reason for the discrepancy? The training methodology of the RF model, including details of hyperparameter optimization, might also be included - possibly in the supplementary material.

10. In the Linear graph-based models subsection, for the external validation results (Figure 5), the performance of the RF models does not appear to be included (unlike for internal validation results in Figure 4). This might be strongly considered.

11. It is then stated that "There is a dependency between the training and internal validation datasets because the same individuals participate in both (although at different timepoints)". If so, it might be clarified as to why the training and internal validation datasets could not have been split at the patient level, such that all data from a patient would appear in either the training or internal validation split, and not both.

12. It might be considered to briefly investigate LR/RF models trained with the top 6 variables by Shapley values (Figure 6), to further support the necessity of causal modelling over other feature importance methods.

Reviewer #2: ROC-AUC values do not have confidence intervals or p-values in the text. Those should be added.

"This re-trained model had a small reduction in AUROC for the COPDGene testing dataset (from 0.75 to 0.73) but resulted in an AUROC of 0.89 when applied to the external SCCOR cohort (Figure 5). This indicates that our model's predictions are robust and generalize beyond COPDGene."

This is not only a much larger AUC than anything I have ever seen WRT prediction of decline in lung function, but is also a highly unusual performance increase from training to test datasets, almost too good as to be unbelievable unless there were some unique attributes in the SCCOR database. This should be addressed somewhere in the discussion.

I'd be curious to know why densitometry or lung volume was not considered as an important variable in the models. It's fine if this happens to be the case, but again it contradicts earlier research and thus there should be something in the discussion address why this happens in this dataset. Other research1 showed that expiratory lung volume and PRMfsad, both of which are available in the COPDGene dataset, were strong predictors of spirometric decline.

1Machine learning for screening of at-risk, mild and moderate COPD patients at risk of FEV1 decline: results from COPDGene and SPIROMICS, Wang et al.

This is my primary issue with the paper and what needs the most investigation/explanation for a revision:

It would be good to plot the difference in FEV1 and FVC distributions between those who were deltaGOLD0=1 vs deltaGOLD0 in the training dataset. Although FEV1 and FVC were not considered in the limited spirometry model, FEF27-75 is highly correlated with FEV1/FVC, indicating that using FEF-25-75 is a proxy for a definition of GOLD 0. It may be that FEF25-75 is such a useful variable because it is highly correlated with FEV1/FVC and therefore is selecting subjects who are just on the border of GOLD0 - GOLD1 classification and therefore likely to convert via noise. This is a problem with predicting change in GOLD status rather than either regression of FEV1 or classification of significant change in FEV1.

The Markov blanket seems to indicate that deltaGOLD0 causes change in FEF, not the other way around. This seems like a major problem with the graph model because the hypothesis is that FEF would predict GOLD progression, correct?

Which Pi10 was used? VIDA, Thirona, or something else?

"Unmeasured inflammation may be the latent confounder between Pi10 and ΔGOLD0 as our model predicts". Doesn't COPDGene have some C-RP measurements? Could you look at those too, at least in a subset?

Reviewer #3: The manuscript "Causal Graph Modelling Identifies Six Factors Linked to Incident COPD" by Gregg et al. is certainly an important advance in the understanding of chronic obstructive pulmonary disease (COPD). In essence, this research identifies relevant variables that influence the decline in lung function. This study includes clinical, genetic and radiological features to identify the predictors of lung function decline. Causal graph modelling techniques were used on a large cohort of 2,643 subjects with a history of smoking but normal spirometry. The analysis identified six key variables that were associated with lung function abnormalities. Among these variables, factors such as "FEF25-75%" and "airway wall thickness" were found to be highly predictive of COPD progression. In addition, the use of predictive models using logistic regression and random forest algorithms added value to the results. An area under the receiver operating characteristic curve (AUROC) value of 0.75 demonstrated the robustness of the model in correctly classifying patients who experience a decline in lung function from those who maintain stable function over a 5-year period. External validation in an independent cohort strengthens the generalisability of the results and provides reasonable support for the reproducibility of the identified predictors in different populations. Importantly, the agnostic approach of this study contributes to the understanding of the heterogeneity of COPD by delineating distinct trajectories of disease progression. The findings pave the way for personalised interventions tailored to specific COPD subtypes. Such insights hold great promise for improving risk stratification, early detection and targeted therapeutic interventions, ultimately improving the management and prognosis of COPD patients. In conclusion, I think the manuscript is well written, well organised and the results seem to be very solid, of great interest to PLoS Medicine readers and very worthy of publication as is.

[LINK]

General journal requests:

---

## [Decision Letter · Decision Letter 2]

9 Jul 2024

Dear Dr. Benos,

Thank you very much for re-submitting your manuscript "Causal Graph Modeling Identifies Six Factors Linked to Incident COPD" (PMEDICINE-D-24-00407R2) for review by PLOS Medicine.

Thank you for your detailed response to the editors' and reviewers' comments. I have discussed the paper with my colleagues, and it has also been seen again by two of the original reviewers. The changes made to the paper were satisfactory to the reviewers. As such, we intend to accept the paper for publication, pending your attention to the remaining editorial comments below in a further revision. When submitting your revised paper, please once again include a detailed point-by-point response to the editorial comments.

[LINK]

In revising the manuscript for further consideration here, please ensure you address the specific points made by each reviewer and the editors. In your rebuttal letter you should indicate your response to the reviewers' and editors' comments and the changes you have made in the manuscript. Please submit a clean version of the paper as the main article file. A version with changes marked must also be uploaded as a marked up manuscript file. Please also check the guidelines for revised papers at http://journals.plos.org/plosmedicine/s/revising-your-manuscript for any that apply to your paper. 

We ask that you submit your revision within 1 week (Jul 16 2024). However, if this deadline is not feasible, please contact me by email, and we can discuss a suitable alternative.

Please do not hesitate to contact me directly with any questions (atosun@plos.org). If you reply directly to this message, please be sure to 'Reply All' so your message comes directly to my inbox.

We look forward to receiving the revised manuscript. 

Sincerely,

Alexandra Tosun, PhD

Associate Editor 

PLOS Medicine

plosmedicine.org

Requests from Editors:

TITLE

Please revise your title according to PLOS Medicine's style. Your title must be nondeclarative and not a question. It should begin with main concept if possible. "Effect of" should be used only if causality can be inferred, i.e., for an RCT. Please place the study design ("A randomized controlled trial," "A retrospective study," "A modelling study," etc.) in the subtitle (ie, after a colon). We also suggest that you spell out "COPD".

FINANCIAL DISCLOSURE

Please state whether any sponsors or funders (other than the named authors) played any role in study design, data collection and analysis, the decision to publish, or preparation of the manuscript. If they had no role in the research, include this sentence: “The funders had no role in study design, data collection and analysis, decision to publish, or preparation of the manuscript.”

DATA AVAILABILITY 

The Data Availability Statement (DAS) requires revision. For each data source used in your study: 

CODE SHARING

We expect all researchers with submissions to PLOS in which author-generated code underpins the findings in the manuscript to make all author-generated code available without restrictions upon publication of the work. In cases where code is central to the manuscript, we may require the code to be made available as a condition of publication. Authors are responsible for ensuring that the code is reusable and well documented. Please make any custom code available, either as part of your data deposition or as a supplementary file. Please add a sentence to your data availability statement regarding any code used in the study, e.g. "The code used in the analysis is available from Github [URL] and archived in Zenodo [DOI link]" Please review our guidelines at https://journals.plos.org/plosmedicine/s/materials-software-and-code-sharing and ensure that your code is shared in a way that follows best practice and facilitates reproducibility and reuse. Because Github depositions can be readily changed or deleted, we encourage you to make a permanent DOI'd copy (e.g. in Zenodo) and provide the URL.

ABSTRACT

1) l.23, please change to: This study aimed...”

2) l.25ff: Please define abbreviations used, such as ‘FEF25-75%’, ‘Pi10’, ‘AUROC’ and ‘FEV1’.

3) l.25ff: Please replace "subject" with participant, patient, individual, or person. Please revise throughout the Abstract and the main text.

4) l.25ff: Please revise the Abstract with regard to the appropriate tense. Methods and Results should be presented in the past tense.

5) l.25ff: Please provide details about the individuals enrolled in the study, such as age, sex, race/ethnicity. Please provide the dates of enrollment (for the COPDGene study). Please also do so for the cohort used for external validation.

6) ll.29-30: Since the six variables identified in your study are part of the main finding, we suggest listing all of them in the Abstract. We also suggest mentioning that you aimed to determine if the variables could predict individuals leaving GOLD 0 status (normal spirometry according to Global Initiative for Obstructive Lung Disease (GOLD) criteria).

7) l.31: Please write ‘yr’ in full.

8) l.31: Please provide the numerical results for the Random Forest predictors.

9) l.33: Please define ‘SCORR’ and include the results (i.e., AUROC) for the external validation.

10) In the last sentence of the Abstract Methods and Findings section, please describe the main limitation(s) of the study's methodology.

AUTHOR SUMMARY

1) Please revise the Author Summary regarding the appropriate tense. For example, ‘Airway wall thickness indicated...’ and/or ‘Methods like graphical models could be used…’.

2) l.41: Please define ‘COPD at first use.

3) l.46ff: Please define all abbreviations the first time you use them or write them out in full. 

INTRODUCTION

l.92, please change to: “In this study, we leveraged…” and revise the following sentences accordingly (i.e. the use of past tense).

METHODS AND RESULTS

We feel that the Results section could be improved in terms of clarity and readability. It is important to guide the reader through the results. In the current format, the connection and introduction of paragraphs sometimes seems a bit disruptive. Please keep in mind that the manuscript should be written in a way that is accessible to a broad audience and to readers who are not familiar with the topic.

1) l.105: Please define ‘CT’ at first use.

2) l.112: We suggest not abbreviating the word "years" with "yrs" and using the full word instead. Please revise throughout the main text.

3) Please include the following statement in the Methods section (We noted it was only included in the ‘SUPPLEMENTAL MATERIALS and METHODS’) and ensure to provide the completed checklist as a Supporting Information file: “This study is reported as per the Transparent Reporting of a Multivariable Prediction Model for Individual Prognosis Or Diagnosis (TRIPOD) statement (S9 Checklist).”

4) ll.207-208: We suggest replacing the subheading with a more general statement, e.g. ‘Variables linked to 5-year ΔGOLD0 progression’. 

5) l.209: As you begin to describe the results here, we feel that it may be confusing to the reader to begin the paragraph with "After identifying..." as it suggests that another results section must have preceded it. Please rephrase.

6) ll.279-280, please revise for clarity.

7) ll.306-307: We suggest replacing the subheading with a more general statement, e.g. ‘Contribution of demographics and SNPs to the prediction of ΔGOLD0 in the random forest model’. 

8) l.309: How was race/ethnicity defined and by whom? We suggest using the term ‘ethnicity’ instead of ‘race’. 

9) l.311, please change to: “…of 65-70 years.”

DISCUSSION

1) Please remove any subheadings from the Discussion.

2) When discussing methods and results, please use the past tense.

3) We suggest more explicit discussion of the limited clinical utility of the model.

TABLES AND FIGURES

1) Table 2: Please ensure that you define all abbreviations used in the table, such as ‘FEV1/FVC’ or ‘AUROC’. Please add a unit for age, e.g. ‘years’.

2) Figure 1/2/3/4/5/6: Please ensure that you define all abbreviations used in the figures, such as ‘FEV1/FVC’ or ‘AUROC’.

3) Figure 3: Please change to ‘Markov Blanket’ (l.622).

4) Figure 4: What does the shaded area in graphs (A) and (D) indicate? 

5) Supplemental Figure S3: Please report exact p-values.

6) Supplemental Figures: Please ensure that you define all abbreviations used in the figures, such as ‘FEV1/FVC’ or ‘AUROC’.

REFERENCES

1) Where website addresses are cited, when specifying the date of access, please use the word “accessed” instead of “cited” (e.g. [accessed: 10/04/2024]).

2) Please ensure that journal name abbreviations match those found in the National Center for Biotechnology Information (NCBI) databases (http://www.ncbi.nlm.nih.gov/nlmcatalog/journals), and are appropriately formatted and capitalised. For example, “New England Journal of Medicine” in reference [3] should be “N Engl J Med”.

SOCIAL MEDIA

To help us extend the reach of your research, please provide any X (formerly known as Twitter) handle(s) that would be appropriate to tag, including your own, your co-authors’, your institution, funder, or lab. Please enter in the submission form any handles you wish to be included when we post about this paper.

Comments from Reviewers:

Reviewer #1: We thank the authors for addressing our previous comments, in particular the additional experiments, and have no further objections pending the resolution of the salient points raised by Reviewer #2.

Reviewer #2: I have carefully read the the article revisions and thank the authors for addressing the main criticisms from me and Reviewer 1. I still think that classifying category changes, especially GOLD, is fraught with issues, but I accept this as an unavoidable limited of the current clinical paradigm

[LINK]

General Editorial Requests

---

## [Editor Report · Decision Letter 3]

17 Jul 2024

Dear Dr. Benos,

Thank you very much for re-submitting your manuscript "Causal Graph Modeling Identifies Six Factors Linked to Incident Chronic Obstructive Pulmonary Disease: a retrospective study" (PMEDICINE-D-24-00407R3) for review by PLOS Medicine.

Thank you for your response to the editors' comments. I have discussed the paper with my colleagues and there remain a few outstanding requests which need to be carefully addressed prior to publication.

[LINK]

We ask that you submit your revision within 1 week (Jul 24 2024). However, if this deadline is not feasible, please contact me by email, and we can discuss a suitable alternative.

Please do not hesitate to contact me directly with any questions (atosun@plos.org). If you reply directly to this message, please be sure to 'Reply All' so your message comes directly to my inbox.

We look forward to receiving the revised manuscript.

Sincerely,

Alexandra Tosun, PhD

Associate Editor 

PLOS Medicine

plosmedicine.org

Requests from Editors:

1) Title: Thank you for revising the title. We suggest changing the title to: Identification of Factors Associated with Incident Chronic Obstructive Pulmonary Disease: A Causal Graph Modeling Study

2) Financial Disclosure: Please include the statement, “The funders had no role in study design, data collection and analysis, decision to publish, or preparation of the manuscript.”, in the Financial Disclosure statement in the online submission form.

3) Data Availability: Please update the Data Availability statement in the online submission form with the details provided in lines 434-437.

4) Abstract: ll.46-50, please change to: The main limitation of the study is that the 5- and 10-year follow-up introduced a mortality bias that disproportionately affected the more severe cases. However, our study focused on spirometrically normal individuals, who have a lower mortality rate. Another limitation is the use of strict criteria to define spirometrically normal individuals, which was unavoidable when studying factors associated with changes in normalized forced expiratory volume in 1 second (FEV1 %predicted) or the ratio of FEV1/FVC (forced vital capacity).

5) Author Summary: In the final bullet point of ‘What Do These Findings Mean?’, please describe the main limitation. Editorial suggestion: The main limitation of the study is that the 5- and 10-year follow-up introduced a mortality bias that disproportionately affected the more severe cases.

6) To prevent additional revisions, please carefully revise the Results sections again regarding the use of the past tense. Examples: “For the “limited spirometry” model, informative variables of ΔGOLD0 included FEF25-75%, demographic measurements (age, height, sex), and CT scan-derived variables (visual severe emphysema and Pi10).” (ll.233-235) or “This indicated that good predictability was not due to data overfit.” (ll.258-259). 

7) ll.265-367, please change to: “To assess whether the COPDGene-trained model generalized to external cohorts, the variables from the "limited spirometry" model without Pi10 were used to retrain the linear model on the COPDGene training set.”

8) l.290, please change to: ”As it is shown in Table 2, FEF25-75% had the most significant coefficient in predicting ΔGOLD0, while age and medium levels of visual CT emphysema did not have significant coefficients.”

9) ll.429-431, please change to: “Overall, this study identified a small subset of variables that may identify individuals at risk for progression to COPD, which could help clinicians develop new disease management protocols.”

10) For Reference [1], please delete “[cited 6 Dec 2020]”.

11) Figure 4: Please clarify whether the shaded area indicates the 95% confidence intervals (i.e., “The shaded areas in the training dataset designate the 95% confidence intervals.”).

12) Supplemental Figures: Please once again ensure that in the figure description, you define all abbreviations used in the supplementary figures. For example: FEV1, FVC, PEF, CT, Pi10 in Supplemental Figure S4.

13) Supplemental Figure S4: Please replace ‘gender’ with ‘sex’.

14) Please provide the completed TRIPOD checklist as a supporting information file. Please use paragraph numbers per section (e.g. "Methods, paragraph 1") for completion (instead of pager numbers).

[LINK]

General Editorial Requests

---

## [Editor Report · Decision Letter 4]

18 Jul 2024

Dear Dr Benos, 

On behalf of my colleagues and the Academic Editor, Jean-Louis Vincent, I am pleased to inform you that we have agreed to publish your manuscript "Identification of Factors Directly Linked to Incident Chronic Obstructive Pulmonary Disease: A Causal Graph Modeling Study" (PMEDICINE-D-24-00407R4) in PLOS Medicine.

I appreciate your thorough responses to reviewers' and editors' comments, and your engagement throughout the editorial process. We look forward to publishing your manuscript, and editorially there is only one remaining minor stylistic point that should be addressed prior to publication. We will carefully check whether the change has been made. If you have any questions or concerns regarding these final requests, please feel free to contact me at atosun@plos.org.

Please see below the minor point that we request you respond to:

1) l.51, please change to: “This study took an agnostic approach…”

Before your manuscript can be formally accepted you will need to complete some formatting changes, which you will receive in a follow up email (including the editorial point above). Please be aware that it may take several days for you to receive this email; during this time no action is required by you. Once you have received these formatting requests, please note that your manuscript will not be scheduled for publication until you have made the required changes.

PRESS

Sincerely, 

Alexandra Tosun, PhD 

Associate Editor 

PLOS Medicine